# Self-Supervised Sentiment Analysis in Spanish to Understand the University Narrative of the Colombian Conflict

Paula Rendón-Cardona [1,*,†], Julian Gil-Gonzalez [2,*,†] 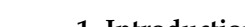, Julián Páez-Valdez [3] and Mauricio Rivera-Henao [1]

1    Faculty of Humanities, Social Sciences, and Education, Universidad Católica de Pereira, Pereira 660005, Colombia; mauricio.rivera@ucp.edu.co
2    Engineering Faculty, Universidad Tecnológica de Pereira, Pereira 660003, Colombia
3    School of Management, Universidad del Rosario, Bogotá 110141, Colombia; juliane.paez@urosario.edu.co
*    Correspondence: paula.rendon@ucp.edu.co (P.R.-C.); jugil@utp.edu.co (J.G.-G.)
†    These authors contributed equally to this work.

**Abstract:** Sentiment analysis is a relevant area in the natural language processing context–(NLP) that allows extracting opinions about different topics such as customer service and political elections. Sentiment analysis is usually carried out through supervised learning approaches and using labeled data. However, obtaining such labels is generally expensive or even infeasible. The above problems can be faced by using models based on self-supervised learning, which aims to deal with various machine learning paradigms in the absence of labels. Accordingly, we propose a self-supervised approach for sentiment analysis in Spanish that comprises a lexicon-based method and a supervised classifier. We test our proposal over three corpora; the first two are labeled datasets, namely, CorpusCine and PaperReviews. Further, we use an unlabeled corpus conformed by news related to the Colombian conflict to understand the university journalistic narrative of the war in Colombia. Obtained results demonstrate that our proposal can deal with sentiment analysis settings in scenarios with unlabeled corpus; in fact, it acquires competitive performance compared with state-of-the-art techniques in partially-labeled datasets.

**Keywords:** self-supervised; sentiment analysis; Colombian conflict; natural language processing

## 1. Introduction

Between 1988 and 2012, Colombia faced an internal armed conflict that plunged the country into a social and political crisis that involved more than 80,000 displaced people, 12,000 violent deaths of community leaders, and at least 5000 attacks on civilian property [1]. In 2016 the final peace agreement was signed; beyond the polarization and bias of public opinion, this negotiation process meant for the victims a transition towards truth, justice, reparation, and non-repetition. The post-conflict is a new stage of recomposition and social justice, which gives way to a society more focused on reconciliation and partial overcoming of that violence. It is the transition, a stage understood as "normalization", where Ávila and Valencia [2] affirm that institutional transformations, social development, and inclusion will be measured as a guarantee for the exercise of rights, reconciliation, and a new culture.

The events unleashed by the war have been reported by large cultural industries and mass media globally, forming an informative agenda that puts the government, the armed forces, the paramilitaries, the guerrillas, and the victims in the daily headlines. It is in this sense that it is necessary to analyze the various media representations of the conflict and the memory of the victims, in particular those of students belonging to the Colombian Network of College Journalism, since they allow evidence of the stories and construction of information and opinion of subjects in professional training, who under their worldviews and narrative logic, permeate the agenda with different perspectives from the hegemonic ones, but in line with what is discussed from other media; This, in the words of Behar and

García [3], occurs because students in their journalistic texts are reconstructing the country's history, complying with the parameters established from the discipline, permeating the agenda of large media.

On the other hand, we recognize a lack of systems that allow extracting information about how the Colombian conflict has been narrated from different media, specifically college journalism. Accordingly, the use of automatic systems for sentiment analysis can be applied to the media narrative aiming to understand the journalism students' opinions about the different conflict stages (war, peace talks, and post-conflict) [4].

Sentiment analysis is a hot topic within the Natural language processing area, its principal objective is to assess peoples' opinions, attitudes, and emotions regarding a specific topic [5]. Sentiment analysis has been utilized in several scenarios, including customer reviews [6], political elections [7], movies reviews [8], and conflict analysis [9]. Regarding the works developed in sentiment analysis, we recognize two main techniques, namely traditional methods and deep learning methods [10].

On the one hand, traditional methods include Lexicon-based approaches and machine learning algorithms. The lexicon-based methods depend on resources composed of words and their corresponding polarity values. Hence, it is possible to evaluate polarity in an unsupervised manner (i.e., without labeled data) [11]; nevertheless, this type of method fails in some tasks mainly because a word can set forth different senses depending on the context [12]. Conversely, in a machine learning context, the sentiment analysis tasks are considered as a binary classification problem (positive and negative) or a multiclass-classification setting (positive, negative, and neutral). Several studies have demonstrated that this type of method can lead to a better sentiment analysis compared with lexicon approaches [13]. However, these approaches require labeled data, which in some scenarios could be infeasible [14].

On the other hand, in recent years, deep learning has shown significant progress in the natural language processing–(NLP) context outperforming traditional NLP approaches. Unlike traditional methods, which use handcrafted features, deep learning models automatically create the required features for the learning process [15]. The most popular deep learning architecture includes Convolutional neural networks–(CNN) [16], Long Short-Term Memory–(LSTM) [17], and Attention-based networks [18]. Still, most deep learning approaches demand a lot of data to guarantee a good generalization performance. The above drawback can be minimized by using transfer learning techniques. Transfer learning uses domain-specific data to fine-tune the pre-trained deep learning models; a common practice is to connect fully connected layers to pre-trained language models (based, for example, on words embedding, CNN, LSTM) [19]. Thereby, the training data is used to compute the parameters belonging to such fully connected layers, while the parameters related to the language model remain untouched [20]. In this regard, the number of trainable parameters is reduced considerably, which gives two crucial benefits: the time spent in the training process is reduced, and it is possible to use small datasets without increasing the probability of overfitting [21]. Nevertheless, despite the transfer learning benefits, the accuracy of sentiment relies on the availability of labeled corpora, and the quality of their labels, which in some practical scenarios can be problematic due to labeling can be tedious, complex, expensive, or even unfeasible [22].

Aiming to tackle sentiment analysis scenarios with limited labels or unlabeled datasets, we recognize two common practices. The first consists of collecting labels from multiple annotators throughout crowd-sourcing platforms such as Amazon Mechanical Turk (https://www.mturk.com/, accessed on 31 March 2021). The attractiveness of these platforms consists in that the labeling processes can be divided into multiple workers obtaining labels (possibly noisy) at a low-cost [22]. Such noisy labels are then used to train a sentiment analysis based on learning from crowds techniques [23]. The second alternative to building sentiment classification systems in presence of limited datasets is to use self-supervised learning methods, which intend to generate pseudo-labels that are used to train a supervised learning model [24]. For sentiment analysis, lexicon-based models are usually

employed combined with filtering stages in order to obtain a set of accurate pseud-labels for training a supervised learning algorithm [14].

Thereby, in this work, we present a self-supervised approach for sentiment analysis in unlabeled Spanish text. The proposed self-supervised methodology is a hybrid methodology that combines lexicon-based and classification methods aiming to build a sentiment analysis system without using any labeled data. Our approach is an extension of SSentiA [14] aiming to deal with Spanish-written texts. Our method comprises three stages. First, it employs a machine translation to build an English version of Spanish corpora. Second, a lexicon-based algorithm is applied to build a set of pseudo-labels with corresponding confidence scores. Finally, a binary classification model is trained with highly-confident pseudo-labels. Such a classifier is used to measure semantic orientation (positive or negative) of the texts whose pseudo-label was categorized with low confidence. We test our proposal over two labeled corpora, namely, CorpusCine and PaperReviews, aiming to validate the performance of our approach in a controlled scenario. Besides, we employ the developed model to assess the polarity in college media that have covered the Colombian conflict, presenting new knowledge about the peace process based on the visibility of the country's college media as a critical scenario of interpretation and as a laboratory of quality production. Achieved results show how our method can predict sentiment polarities from unlabeled or partially labeled data.

The remainder is organized as follows. Section 2 exposes the related work and main contributions of the proposal. Section 3 describes the methodology. Sections 4 and 5 present the experiments and discuss the results. Finally, Section 6 outlines the conclusions and future work.

## 2. Related Work

Sentiment analysis is a hot topic in the natural language processing context that comprises the analysis of people's opinions and sentiments from written language [9]. Sentiment analysis has successfully applied to different scenarios such as customer opinions [25], enterprise marketing [26], and public opinion about local and international political conflicts [5]. In the literature, we recognize several works regarding political disputes. For example, authors in [27] perform a sentiment analysis from tweets during the 2016 primary debates aiming to assess the opinions' polarity about Donald Trump. Alike, in [28] a sentiment analysis is employed to carry out a bias detection in Western news about the Palestinian/Israeli conflict. On the other hand, we recognize additional sentiment analysis approaches to analyze conflicts such as the Iran–USA nuclear deal [29], the USA–China trade war [30,31], and the problem of Syrian refugees [9]. Conversely, concerning the Colombian conflict, we notice a lack of sentiment analysis-based studies to describe such conflict and its stages (Conflict, peace agreement, and post-conflict). We only identified two relevant works. The first is the proposed in [32], which uses a crawled dataset from Twitter to analyze the perception of the post-conflict in Colombia from two perspectives: local and foreign perception. Besides, [33] uses a qualitative approach to determine the semantic orientation of information captured from Twitter, aiming to evaluate Norway's contribution to the peace process in Colombia.

On the other hand, we broadly identified three methods for measuring semantic orientation, namely, lexicon-based models, supervised learning methods, and deep learning approaches. Lexicon-based methods comprise the use of a dictionary, where each word is labeled as positive or negative. Then, a sentence is tokenized, and it is matched with the words in the dictionary, aiming to compute the sentiment polarity [34]. However, one of the main drawbacks of lexicon-based methods is that applying one or another dictionary could lead to different polarity assessments [35]. In contrast to lexicon-based methods, machine learning-based approaches use manually labeled training data to code relevant patterns related to the sentiment analysis. Supervised learning methods usually outperform dictionary-based systems, and in fact, several relevant sentiment analysis models are built from machine learning algorithms [36]. Unlike previous methods, which should spend

considerable time defining handcrafted features and performing feature engineering, deep learning models automatically perform such tasks [15]. Moreover, deep learning has exhibited important advances in several areas, including Computer Vision [37] and NLP, where the application of deep learning to sentiment analysis has become popular [38]. However, one of the main drawbacks of these methods (machine learning and deep learning) for sentiment analysis is the requirement of a labeled dataset, where the actual label (gold standard) is provided for each document in the corpus. In different scenarios, collecting such true labels is a non-trivial task because it can be tedious, expensive, or even unfeasible [39].

Several self-supervised approaches have been presented to deal with sentiment analysis problems with no labels. Such methods are mainly based on the combination of a lexicon and machine learning algorithms [40,41]. Accordingly, we notice that most of the works related to self-supervised learning for sentiment analysis are developed for documents written in the English language. However, the majority of documents related to the Colombian conflict are Spanish-written. Spanish is one of the most spoken languages in the world (more than 500 million speakers), just behind English, Chinese, and Hindi [42]. However, despite the importance of the Spanish language, we identify that the area of sentiment analysis in the Spanish language is not as well-resourced as in the English language [43].

According to the related previously, we propose a self-supervised approach to face sentiment analysis without labels. Our approach is an extension of the work in [14], so-called SSentiA. Similar to SSentiA, our proposal uses a combination of a lexicon and supervised learning-based methods. However, unlike SSentiA, we employ an additional step (Machine-translation) to deal with Spanish-written texts. Besides, our proposal shares some similarities with the work in [44] because we use a set of basis labels to train a sentiment analysis model, which is then employed to assign labels to unlabeled texts. However, while the basis labels in [44] are given by an expert (Weakly Supervised), our approach does not receive any information about the gold standard, and such basis labels are generated in an unsupervised way, which configures a more challenging scenario.

Our proposal is applied over a corpus composed of documents extracted from college media to understand the Colombian conflict's university narrative. To the best of our knowledge, this is the first attempt to analyze sentiment polarity speech from the university perspective. Previous works [32,33] have performed similar analyses, but they use information extracted from Twitter.

## 3. A Self-Supervised Learning Approach for Sentiment Analysis

A machine learning-based approach for sentiment analysis comprises computing a function that assigns a binary number to each document or sentence, where $-1$ represents a negative polarity, whereas 1 stands for a positive document. Such a function $f$ is estimated by using a training dataset with $N$ samples $\mathcal{D} = \{x_i, y_i\}_{i=1}^{N}$, where $x_i$ is the $i$-th document, and $y_i \in \{-1, 1\}$ is the polarity (gold standard) for document $x_i$. However, such a gold standard is not available or scarce in many scenarios. Thereby, it is necessary to employ approaches that allow building sentiment analysis frameworks in this type of scenario.

Accordingly, in this work, we employ a self-supervised approach based on the Self-supervised Sentiment Analyzer for classification from unlabeled data–(SSentiA) proposed by [14]. Such an approach generates pseudo-labels using a lexicon-based method; then, these labels are enhanced using a supervised classification scheme. Below, we describe the steps related to our proposal.

### 3.1. Machine-Translated Corpus

As we have pointed out previously, the number of resources for Spanish sentiment analysis is limited; specifically, we recognize a lack of dictionaries to apply lexicon-based strategies. Conversely, for the English language, there is a considerable quantity of dictionaries for sentiment analysis, including domain-specific dictionaries [34]. Accordingly, following the ideas in [11], we used a machine translation (precisely Google translate)

in order to build an English version $\tilde{X}$ of the original corpus $X = \{x_i\}_{i=1}^N$. The above allows us to take a profit from English dictionaries.

### 3.2. Lexicon-Based Label Generation

The second step in our approach comprises the generation of highly accurate pseudo-labels to feed a supervised learning classifier. For specific testing, we employ the method LRSentia [14]. LRSentia is a lexicon-based method that allows assessing the semantic orientation of a review in an unsupervised manner (i.e., without labels).

Given the translated corpus $\tilde{X} = \{\tilde{x}_i\}_{i=1}^N$, LRSentiA divide each translated document $\tilde{x}_i$ into sentences. Accordingly, we have a set $S_i = \{s_{1,i}, \ldots, s_{R_i,i}\}$ containing the sentences conforming the document $\tilde{x}_i$, where $R_i$ is the number of sentences in document $\tilde{x}_i$. Then, a pre-processing algorithm is applied over each sentence in order to exclude words that do not have significant relevance for the sentence polarity (for more details, see [11]). Once we have performed the pre-processing step, the sentiment polarity of each sentence $P(s_{j,i})$ with $j \in \{1, \ldots R_i\}$ is computed.

Given the polarity for all the sentences, $P(s_{1,i}), \ldots, P(s_{R_i,i})$, the overall polarity for document $\tilde{x}_i$ is estimated as $P(S_i) = \sum_{j=i}^{R_i} s_{j,i}$. Therefore, the semantic orientation for the $i$-th document is given as

$$P(x_i) = \begin{cases} 1 \ (\text{Positive}), & P(S_i) \geq 0 \\ -1 \ (\text{Negative}), & P(S_i) < 0 \end{cases}. \tag{1}$$

Besides to the semantic orientation, LRSentiA provides the confidence score $\rho_i$ for the predictions $P(x_i)$. The confidence score for the document $x_i$ is estimated as follows:

$$\rho_i = \frac{P(S_i)}{\displaystyle\sum_{j=1}^{J} \text{abs}(P(s_{j,i}))}, \tag{2}$$

where we recall that $P(S_i) = \sum_{j=i}^{R_i} s_{j,i}$. As the Equation (2) indicates, the confidence score of review $i$, depends on the difference between the positive ($s_{j,i} > 0$) and negative ($s_{j,i} < 0$) polarities; hence, the presence of highly polarized documents points out a high confidence score. In summary, documents with high confidence scores are supposed to be less prone to misclassification and can be used as pseudo-labels to train a supervised learning algorithm [11].

On the other hand, we define the vector $\rho \in \mathbb{R}^N$ containing the confidence scores for the documents in the corpus. Based on such vector, the confidence group $\zeta_i$ for the $i$-th document is defined as

$$\zeta_i = \begin{cases} \text{Very-high}, & \rho_i > \delta \\ \text{High}, & \delta - 0.5\sigma_\rho < \rho_i \leq \delta \\ \text{Low}, & \delta - \sigma_\rho < \rho_i \leq \delta - 0.5\sigma_\rho, \\ \text{Very-low}, & 0 < \rho_i \leq \delta - \sigma_\rho \\ \text{Zero}, & \rho_i = 0 \end{cases} \tag{3}$$

where $\delta$ is the threshold used to define the confidence categories, which is computed as, $\delta = \mu_\rho + \sigma_\rho$, being $\mu_\rho$, and $\sigma_\rho$ respectively the mean and standard deviation of vector $\rho$. The categorization shown in Equation (3) is intended to minimize incorrect predictions while maximizing the number of samples to train the supervised learning algorithm. Previous studies [11,14] have shown the presence of a correlation between the confidence scores and the prediction accuracies.

### 3.3. Machine Learning Classifier

As we have established below, there is a relation between the prediction accuracy and the confidence scores; the higher the confidence scores, the higher the accuracy. Accordingly, let $\hat{y}_1 \in \{-1, 1\}^{N_1}$ a pseudo-labels vector containing the lexicon-based predictions whose confidence scores lies in the categories "Very-high" and "High" (see Equation (3)). We employ such pseudo-labels and their corresponding documents to train a supervised classification model. Then, the trained model is used to predict the semantic orientation $\hat{y}_2 \in \{-1, 1\}^{N_2}$ of documents whose lexicon-based label had "Low", "Very-low", and "Zero" confidence. Finally, the overall prediction $\hat{y} \in \{-1, 1\}^N$ is composed from the concatenation of pseudo-labels $\hat{y}_1$, and the predictions $\hat{y}_2$, being $N = N_1 + N_2$. As in [14], each document is parametrized using unigram and bigram-based term frequency–inverse document frequency (tf–idf).

## 4. Experimental Set-Up

### 4.1. Datasets

The presented methodology is tested over three corpora. The first two, CorpusCine and PaperReviews, configure a controlled experiment because, for both corpora, we have access to the ground truth (the actual semantic orientations). Conversely, a third corpus related to the Colombian conflict is employed, representing a more challenging scenario because such a dataset has not been labeled.

#### 4.1.1. Labeled Datasets

CorpusCine is a dataset formed by 3878 Spanish-written movie reviews captured from the MuchoCine website (https://muchocine.net/, accessed on 31 March 2021). Each document is rated using an integer tag ranging from 1 (unpleasant movie) to 5 (excellent movie). The rating distribution for the CorpusCine dataset is presented in Table 1. We remark that using such a dataset for sentiment analysis configures a multi-class classification problem; however, as we have clarified in Section 3, our sentiment analysis approach was designed for binary settings (positive and negative documents). Accordingly, we use the procedure in [45] aiming to convert the CorpusCine into a binary problem, as follows. Documents with a rating of one or two are considered a negative review ($-1$); conversely, manuscripts with a label of four and five are estimated as positives (1). We discard documents with a tag of three because they correspond to neutral or mixed opinions [46]. This dataset is publicly available (http://www.lsi.us.es/~fermin/corpusCine.zip, accessed on 31 March 2021).

**Table 1.** Rating distribution for the labeled corpora.

| Rating | Number of Documents | |
|:---:|:---:|:---:|
| | **CorpusCine** | **PaperReview** |
| 1 | 351 | 36 |
| 2 | 923 | 136 |
| 3 | 1253 | 104 |
| 4 | 890 | 91 |
| 5 | 461 | 15 |

On the other hand, we use the PaperReviews corpus, which comprises paper reviews sent to an international conference in Spanish. The corpus has a total of 382 documents labeled using a five-point scale, indicating the opinion about the paper quality. Table 1 shows the rating distribution for the PaperReviews dataset. Alike for CorpusCine, we use a methodology to treat PaperReviews as a binary classification problem. Samples with ratings of one or two are considered negative reviews; similarly, documents with ratings of four or five are categorized as positive reviews. PaperReviews is publicly available (https://archive.ics.uci.edu/ml/datasets/Paper+Reviews, accessed on 31 March 2021).

4.1.2. Colombian Conflict Dataset

The third dataset used in this work comes from the Universidad Católica de Pereira. It comprises a collection of journalistic articles written by young university students in Colombia, which is a product of the project: Analytical Center of University Cultural Productions in the Context of the Conflict (caPAZ), funded by the Ministry of Science, Technology, and Innovation (Minciencias) and the National Center for Historical Memory (CNM) of Colombia (under the code: 1349-872-76354, agreement 872 of 2020). This corpus includes news written by the 24 college media of the Colombian Network of College Journalism from 2001 to 2021. The dataset includes digital, printed, sound, and audiovisual news, for 3049 news items and 3,451,486 words related to the armed conflict, the memory of the victims, and the peace process in Colombia. For the case of this article, we will focus on the news articles that appeared on the web pages of the 24 college media because they gave us greater diversity, greater reliability in the extraction process, and a more significant impact due to their high production. However, we only used the news articles from 8 college media since they were the ones with more than 50 news items. In conclusion, a total of 2373 digital news articles were analyzed, see Table 2. These news items were collected through a web-scraping technique, using three lemmatized keywords (conflicto armado, memoria de las víctimas, and proceso de paz), with the aim of identifying these regular expressions in the logical operators that run through the HTML structure of each Web page. Likewise, a web-crawling technique was used to explore the addressing links of each page and build a relevant relationship graph. The 2373 news items were converted into raw text format .txt. Finally, the documents' semantic orientations are generated by performing a lexicometry analysis using the Alceste–Reinert method for textual data clustering [47]. Such dataset is publicly available (https://zenodo.org/record/6384840#.Yj5Av-fMJPZ, accessed on 31 March 2021).

**Table 2.** Number of news per media in the Colombian conflict corpus.

| Media | N° News |
|---|---|
| Universidad de Ibagué | 89 |
| Universidad del Valle | 60 |
| Universidad de la Sabana | 281 |
| Uniminuto | 508 |
| Jorge Tadeo Lozano | 800 |
| Santiago de Cali | 94 |
| Universidad del Rosario | 175 |
| Universidad Autónoma de Bucaramanga | 366 |

The Alceste–Reinert method focuses on the identification of lexical worlds. This method makes it possible, through statistical analysis, to identify the appearance of specific forms in the content of a speech. Reinert's main thesis [48] asserts that all discourse expresses a system of lexical words, which organize thought and give coherence to what is described by the enunciators. These lexical worlds are identified through top-down hierarchical clustering exercises and expressed in textual dendrograms, namely, a tree-shaped diagram to represent the hierarchy of categories according to the degree of similarity and shared characteristics. For the case of this research, textual dendrograms were generated for each medium analyzed. These eight dendrograms compile the 2373 news items. The labeling process consisted of qualitative analysis through direct observation of each textual dendrogram. Polarity coding was performed by an investigator, who labeled the material as positive and negative. The assignment of the said label was made with each percentage community expressed in the dendrograms. It is essential to clarify that the labeling process was done for each analyzed college media and not for each news item in the corpus.

*4.2. Methodology Training and Validation*

The proposed self-supervised learning approach combines the lexicon-based method LRSentiA with supervised learning techniques. LRSentiA generates pseudo-labels with their corresponding confidence scores. Then the machine learning approaches capture critical patterns on the data with the highest confidence scores aiming to enhance the polarity representation of complex documents where the LRSentiA's performance is poor [14]. Accordingly, for the LRSentiA method, we use the opinion lexicon curated by authors in [49], which comprises 4783 negative and 2006 polarity words. On the other hand, for the machine learning stage, each document is parametrized using the well-known unigram and bigram-based tf–idf (term frequency-inverse document frequency) scores. Besides, we test four binary classification algorithms, namely, Logistic Regression (LR), Support vector machines (SVM), Random Forest (RF), and Naive–Bayes (NB).

Aiming to validate the efficacy of our self-supervised approach, we use the procedure in Figure 1. We employ two state-of-the-art lexicons for English-written texts, namely, VADER [50], TextBlob (https://textblob.readthedocs.io/en/dev/, accessed on 31 March 2021). Moreover, we use a publicly available Spanish lexicon (https://www.kaggle.com/rtatman/sentiment-lexicons-for-81-languages, accessed on 31 March 2021). On the other hand, the quality assessment is performed by computing overall accuracy and the F1 score between the predicted labels and the ground truth. The Python codes for our method are publicly available (https://github.com/juliangilg/SSentiA_Sp, accessed on 31 March 2021).

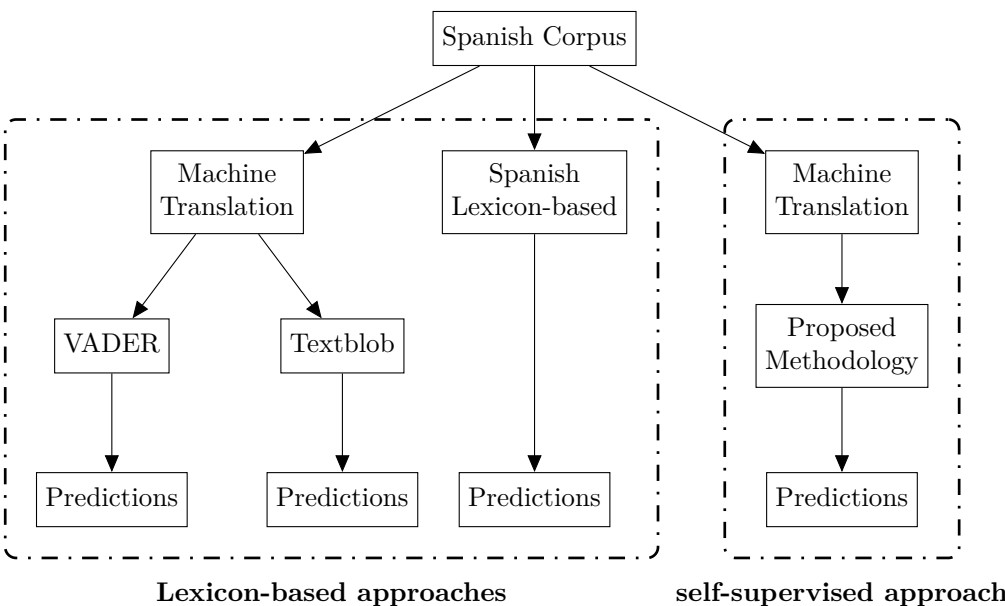

**Figure 1.** Validation procedure using some state-of-the-art lexicon-based methods.

## 5. Results and Discussion

*5.1. Labeled Datasets*

As we established in Section 3.1, we use a machine-translation approach to take advantage of the English resources; thus, it is crucial to verify that the documents' semantic orientation is not affected by the translation process [51,52]. In this regard, we perform an initial experiment aiming to evaluate the machine-translation impact for sentiment analysis. For such an experiment, we use the CorpusCine and PaperReviews that review dataset, each dataset since these corpora contain the actual labels. We build two sentiment analysis systems (based on machine learning algorithms) for each corpus. The first uses the documents in their original language (Spanish), while the second employs an English-translated version of such documents. Each document is parametrized by using unigram and bigram-based term frequency-inverse document frequency. We use a five-fold validation scheme, where the performance is assessed in terms of the overall accuracy. Table 3 shows a comparison of

a machine learning-based model for sentiment analysis trained by using both original and translated corpora. From Table 3, we remark that there are no significant differences between the performance (in terms of overall accuracy) of sentiment analysis systems from Spanish and English-translated corpora. Consequently, we experimentally demonstrate that the sentiment polarity is preserved after the English translation in most cases.

**Table 3.** Comparison of sentiment analysis performance (in terms of overall accuracy) between Spanish corpora and English-translated corpora. LR: logistic regression, SVM: support vector machines.

| | Original Corpora (Spanish) | | Machine-Translated Corpora (English) | |
|---|---|---|---|---|
| | **LR** | **SVM** | **LR** | **SVM** |
| CorpusCine | $82.30 \pm 1.01$ | $83.62 \pm 1.06$ | $81.70 \pm 1.24$ | $83.40 \pm 0.51$ |
| PaperReviews | $78.09 \pm 1.01$ | $79.52 \pm 2.54$ | $77.61 \pm 2.04$ | $76.66 \pm 2.07$ |

Second, we perform a controlled experiment aiming to verify the capability of our proposal to compute the semantic orientation in the absence of labels. We use the CorpusCine and PaperReviews reviews dataset described previously. We remark that these datasets contain the true semantic orientations; nevertheless, such labels are only used to compute performance metrics.

As we pointed out in Section 3.2 our approach uses a lexicon-based algorithm (LRSentiA) to generate pseudo-labels that are then used to train a machine learning model. Besides, LRSentiA computes the confidence score for each pseudo-label Equation (2), where it has been established that there exists a relation between such confidence scores and the prediction accuracy [14]. Accordingly, we perform a first experiment aiming to confirm previous affirmation experimentally. Table 4 shows the prediction accuracy compared with the confidence groups defined in Equation (3).

**Table 4.** Prediction accuracy of LRSentiA for the defined confidence groups (Equation (3)) in a machine-translated version of CorpusCine and PaperReview datasets.

| | Confidence Group | Confidence Score $\rho$ Equation (2) | Accuracy % | # Reviews |
|---|---|---|---|---|
| | Very-high | $(0.33, 1.00]$ | 74.23 | 788 |
| | High | $(0.24, 0.33]$ | 68.23 | 318 |
| CorpusCine | Low | $(0.14, 0.24]$ | 61.90 | 504 |
| | Very-low | $(0.01, 0.14]$ | 58.06 | 856 |
| | Zero | $(0.00, 0.01]$ | 47.16 | 159 |
| | Very-high | $(0.69, 1.00]$ | 63.04 | 92 |
| | High | $(0.51, 0.69]$ | 48.27 | 29 |
| PaperReviews | Low | $(0.34, 0.51]$ | 46.66 | 45 |
| | Very-low | $(0.01, 0.34]$ | 51.25 | 80 |
| | Zero | $(0.00, 0.01]$ | 28.15 | 32 |

From Table 4, it is possible to notice that, for both labeled datasets, there exists a significant relationship between the confidence scores and the prediction accuracy, the Very-high confidence group is related to the highest prediction accuracy. In contrast, the lowest accuracy corresponds to the Zero confidence group. We compute the Pearson coefficient between the prediction accuracy and the mean confidence score in each group to support our observations. Hence, we obtain a value of 0.8853 for CorpusCine and 0.8999 PaperReviews, which indicates a strong linear relationship between the analyzed variables confirming our initial qualitative analysis.

As a second experiment, we use lexicon-based algorithms for the CorpusCine dataset. We employ the well-known English dictionaries TextBlob and VADER, applied over an English version of the studied dataset. Further, we use the Spanish dictionary described

in Section 4.2. From Table 5, we observe that CorpusCine, in terms of the F1 score, the dictionary TextBlob obtains the best performance. On the other hand, regarding PaperReview, we note that the VADER lexicon outperforms all its competitors. However, we observe that the lexicon-based methods exhibit a considerably low-performance in general terms. The above behavior is explained in the sense that dictionaries-based algorithms obtain their best yielding with highly polarity text; however, they are ineffective in assessing the sentiment orientation of cases with mixed opinions [14].

**Table 5.** Performance of lexicon-based approaches for CorpusCine and PaperReview datasets and their machine-translated English versions. For the Spanish lexicon, we use the available in https://www.kaggle.com/general/88685 (accessed on 21 March 2021). The best performance is highlighted in bold.

|  | Corpus Language | Lexicon | Precision | Recall | F1 Score | Accuracy |
|---|---|---|---|---|---|---|
| CorpusCine | English (translated) | TextBlob | **0.6975** | 0.5828 | **0.6350** | **0.5939** |
|  |  | VADER | 0.5966 | 0.5685 | 0.5822 | 0.5764 |
|  | Spanish | Spanish lexicon | 0.5500 | **0.6084** | 0.5900 | 0.5649 |
| PaperReviews | English (translated) | TextBlob | 0.6645 | 0.6062 | 0.6340 | 0.5271 |
|  |  | VADER | **0.6918** | **0.6628** | **0.6770** | **0.6007** |
|  | Spanish | Spanish lexicon | 0.6382 | 0.6337 | 0.6359 | 0.5921 |

Finally, we test our hybrid proposal over the labeled datasets. Table 6 shows the performance of our approach compared with different methods (Unsupervised, supervised, and hybrid) to perform sentiment analysis in Spanish corpora. Analyzing the results for CorpusCine in Table 6, we first evidence that our proposal reaches the best performance by using a logistic regression classifier, which is remarkable due to it can only deal with linear-separable data. The above suggests a linear structure may exist in the features extracted from the documents. On the other hand, we note that the approaches with the worst performance are those based on unsupervised learning algorithms, which is not surprising. As it was argued below, such types of methods cannot measure documents with mixed opinions. Now, concerning the supervised learning methods, we note that the F1 score considerably outperforms unsupervised algorithms. Such behavior is due to supervised models extracting relevant patterns related to the sentiment orientation. Then, we remark that the works based on hybrid approaches (i.e., combining supervised and unsupervised techniques) [53,54] achieve the best classification scores (F1 score, Accuracy). However, regarding our hybrid approach, we observe a significantly lower performance; in fact, it only outperforms the unsupervised models. To explain such an outcome, we highlight that unlike the approaches in [53,54], our proposal does not use any information about the gold standard. Conversely, it generates pseudo-labels using the lexicon-based method LRSentiA, which is then used to train a supervised classifier. Therefore, we argue that the performance of our approach can be increased by providing limited labeled data.

**Table 6.** Performance of our hybrid approach using four different classification algorithms and compared with state-of-the-art approaches. LR: Logistic Regression, SVM: Support Vector Machine, RF: Random Forest, NB: Naive Bayes.

|  | Source | Approach | Labels? | Precision | Recall | F1 Score | Accuracy |
|---|---|---|---|---|---|---|---|
| CorpusCine | [55] | Unsupervised | ✗ | 0.6393 | 0.6274 | 0.6333 | 0.6316 |
|  | [56] | Supervised | ✓ | 0.8721 | 0.8701 | 0.8710 | 0.8708 |
|  | [53] | Supervised | ✓ | - | - | - | 0.7713 |
|  | [57] | Supervised | ✓ | 0.8531 | 0.8959 | 0.8739 | 0.8667 |
|  | [53] | Hybrid | ✓ | - | - | - | 0.8086 |
|  | [54] | Hybrid | ✓ | 0.8858 | 0.8857 | 0.8856 | 0.8857 |
|  | Proposal (LR) | Hybrid | ✗ | 0.6727 | 0.6632 | 0.6679 | 0.6664 |
|  | Proposal (SVM) | Hybrid | ✗ | 0.6624 | 0.6509 | 0.6566 | 0.6546 |
|  | Proposal (RF) | Hybrid | ✗ | 0.6484 | 0.5959 | 0.6210 | 0.6042 |
|  | Proposal (NB) | Hybrid | ✗ | 0.6384 | 0.5744 | 0.6047 | 0.5840 |
| PaperReviews | [58] | Supervised | ✓ | 0.4621 | 0.3596 | - | 0.3600 |
|  | [59] | Supervised | ✓ | 0.7600 | 0.6600 | 0.7100 | 0.7400 |
|  | [60] | Hybrid | ✓ | 0.7100 | 0.7200 | 0.7100 | 0.7100 |
|  | Proposal (LR) | Hybrid | ✗ | 0.695 | 0.5407 | 0.6082 | 0.4212 |
|  | Proposal (SVM) | Hybrid | ✗ | 0.6942 | 0.5378 | 0.6061 | 0.4176 |
|  | Proposal (RF) | Hybrid | ✗ | 0.6942 | 0.5378 | 0.6061 | 0.4176 |
|  | Proposal (NB) | Hybrid | ✗ | 0.6942 | 0.5378 | 0.6061 | 0.4176 |

Conversely, regarding the results (Table 6), we notice that, similar to previous outcomes, the best F1 scores come from supervised and hybrid approaches. However, we remark a considerable low yielding from our proposal and the deep learning approach introduced in [58]; in fact, both methods are defeated by the lexicon-based algorithms (see Table 5). Concerning the deep learning-based results, we argue that such low performance can be caused by a lack of generalization (overfitting). Now with respect to our hybrid proposal, we explain the undesired results in two regards. First, like the results for the CorpusCine dataset, the training step does not have access to the actual labels; therefore, the behavior of our approach can be enhanced by using partially labeled corpora. Second, we observe that the results from our method are skewed towards positive labels (the precision is greater than recall), which is caused for a discrepancy between the paper is evaluated and the way the review is written [60].

Finally, aiming to evaluate the behavior of our hybrid proposal in scenarios with limited labeled data, we carry out an additional experiment where we vary the number of labels. Let be $p \in [0, 1]$ the ratio of available labeled data. We split the CorpusCine review dataset into two groups. The first, $X_l$ is conformed by $pN$ (being $N$ the number of documents in such corpus) samples with their corresponding true labels $y_l \in \{0, 1\}^{pN}$; conversely, the second group $X_u$ contains $N(1 - p)$ unlabeled documents. Hence, we apply the LRSentiA algorithm over the unlabeled corpus $X_u$ to generate pseudo-samples. LRSentiA classifies such labels into five groups (Very-high, High, Low, Very-low, and zero) according to their confidence. Then, we conform the sets $X_t$, and $y_t$ by concatenating the samples and labels from the Very-high and High categories together with the labeled data ($X_l, y_l$). $X_t$ and $y_t$ is then used to feed a classification algorithm to predict the label for documents belonging to low, very-low, and zero confidence. For each value of $p$, we use a five-fold validation scheme, where the sets $X_l$, and $X_u$ are conformed randomly. The performance is measured in terms of the overall accuracy. Figure 2 shows the mean and standard deviation for the accuracy of our proposal as a function of the ratio of labeled data. For specific testing, we compare a logistic regression classifier and a Support vector machine since they achieved the highest performances in previous experiments (see Table 6). From Figure 2, we remark that in most cases, the performance of our approach increases as there are more labeled data. Besides, we highlight that with the 40% of labeled data, our

proposal achieves an accuracy near 0.8 for CorpusCine data and near 0.7 for PaperReview dataset, which remarkably is comparable with state-of-the-art results.

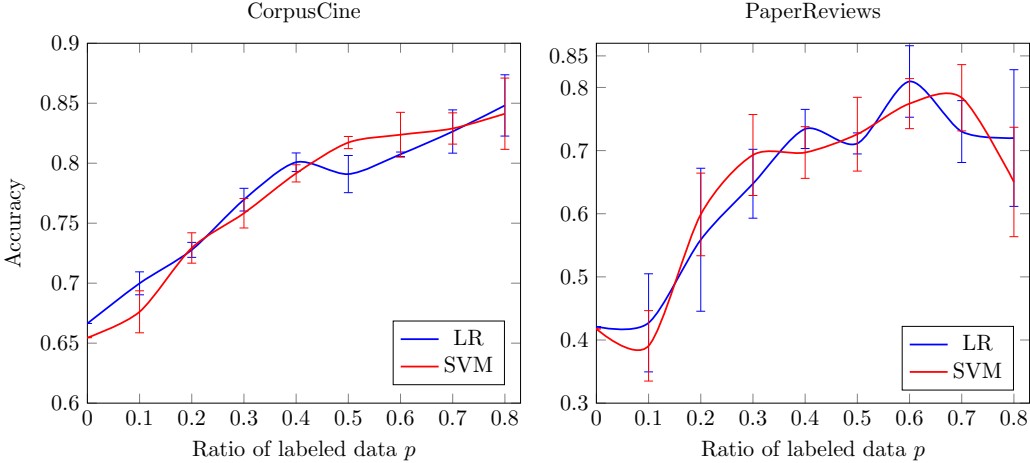

**Figure 2.** Accuracy of our approach (LR and SVM) as a function of the number of labeled data.

### 5.2. Colombian Conflict Dataset

As a final experiment, we apply our proposal to each document in the Colombian conflict dataset aiming to understand the narrative of the war from the journalism students' perspective. Aiming to validate the obtained results, we use the qualitative labels explained in Section 4.1.2. However, we note that such labels are not given for each document but for each media. Accordingly, to perform a fair comparison, we use the following procedure. Let $X_m = \{x_1^{(m)}, \ldots, x_{N_m}^{(m)}\}$ a set containing $N_m$ documents corresponding to the $m$-th media. We use our hybrid method to obtain the sentiment polarity $\boldsymbol{y}_m = [y_1^{(m)}, \ldots, y_{N_m}^{(m)}]$ for each document in $X_m$. Accordingly, the polarity of $m$-th media is computed as:

$$P(X_m) = \begin{cases} 1, & \sum_{i=1}^{N_m} y_1^{(m)} \geq 0 \\ 0, & \text{In other case} \end{cases}. \tag{4}$$

Table 7 shows a comparison between the semantic orientation obtained by using our approach under four configurations (employing four classification algorithms: LR, SVM, RF, and NB) and a set of qualitative labels, which are obtained using the identification of lexical worlds [47] (see Section 4.1.2).

**Table 7.** Sentiment analysis for the Colombian conflict dataset. We compare the behavior of our proposal with qualitative labels for each media. For our approach, we present the predicted polarity and the ratio of documents whit positive semantic orientation.

| Media | Qualitative Polarity | Proposal | | | |
|---|---|---|---|---|---|
| | | **RL** | **SVM** | **RF** | **NB** |
| Universidad de Ibagué | P | P(0.8000) | P(0.7888) | P(0.7666) | P(0.8333) |
| Universidad del Valle | P | N(0.1016) | N(0.1016) | N(0.1016) | N(0.1016) |
| Universidad de la Sabana | P | P(0.7651) | P(0.8066) | P(0.8259) | P(0.8701) |
| Uniminuto | P | P(0.7199) | P(0.7317) | P(0.8579) | P(0.8086) |
| Jorge Tadeo Lozano | P | P(0.7101) | P(0.8412) | P(0.8941) | P(0.8981) |
| Santiago de Cali | P | N(0.4772) | N(0.4512) | N(0.4610) | N(0.4090) |
| Universidad del Rosario | P | P(0.5614) | P(0.6359) | P(0.6052) | P(0.6403) |
| Universidad Autónoma de Bucaramanga | P | P(0.5390) | P(0.5306) | P(0.6170) | P(0.7142) |

From Table 7, we first notice that according to the qualitative assessment, all of the considered media expresses positive aspects of the Colombian conflict and its stages. Now, regarding the result of our self-supervised approach, we note a similar tendency, given that most of the media are categorized as positives; such outcomes are consistent when we vary the classification scheme (RL, SVM, RF, and NB), obtaining an overall accuracy of 75%. Specifically, we remark that for media Universidad del Valle and Santiago de Cali the sentiment orientation predicted for our approach differs from the qualitative labels; hence, to explain such outcomes, we analyze each media individually. For the Universidad del Valle media, we observe it contains 60 documents, and only 26 were categorized as very-high or high confidence. The above indicates that the supervised learning approach was trained with 26 samples, which indicates that the overfitting causes the results obtained. Conversely, regarding the Santiago de Cali media, we notice that despite the predicted label being negative, the ratio of documents with a positive semantic orientation is close to 0.5, which indicates the robustness of our approach to recognizing the polarity of documents related to the Colombian conflict.

On the other hand, the results are complemented by the lexicometric approach. College journalism narrated the conflict with a positive stance; this is reflected in the need to understand Colombia's conflict and propose solutions and scenarios of agreement. From journalism, we could affirm that college journalism focused on presenting future actions and not falling into the traditional narrative that was merely informative.

## 6. Conclusions

In this paper, we study the problem of identifying sentiment polarity from unlabeled corpora. The presented approach is based on the so-called SSentiA [14], which is built from a hybrid method that combines lexicon-based algorithms and supervised learning models. Unlike SSentiA, our proposal uses a machine translation approach to perform sentiment analysis in Spanish-written documents. Our strategy was first tested using the CineCorpus, and PaperReview datasets, which configure a controlled scenario because, for such corpora, we have access to the actual labels (polarities). According to the results, we observe that the proposal's performance is considerably lower compared with state-of-the-art supervised learning and hybrid approaches, which use the gold standard. However, we highlight that when our methodology has access to limited labeled data (around 40%), it can perform similarly to methods with the fully labeled dataset. On the other hand, we tested the proposal in a more challenging corpus related to the Colombian conflict. The outcomes were compared with qualitative labels based on the identification of lexical worlds [47], showing an overall accuracy of 75%. From the journalism perspective, we can conclude that the two approaches used (quantitative and qualitative) had similar results and evidenced the positive polarity of the university journalistic story. The qualitative approach, based on the Alceste–Reinert lexicometric method, allowed us to identify the polarity of the discourse, and also the journalistic themes worked by the media, for this reason, we can conclude affirming that the Colombian university students narrated the conflict positively from 3 topics: the armed conflict, the memory of the victims and the peace process.

In future work, the authors plan to follow the work in [5] in order to combine the developed methodology with topic modeling such as the Latent Dirichlet Allocation (LDA), aiming to extract the relationship between the sentiment polarities and the most relevant topics in the documents to capture more relevant insights about the development of the Colombian conflict from the perspective of journalism students. On the other hand, we observe that our approach is considerably affected in the presence of corpora with few documents and for class imbalance; accordingly, we plan to use data manipulation techniques such as weighting data examples or data augmentation [59,61] aiming to improve the performance of our approach. Finally, in this work, we use a traditional approach (based on machine learning algorithms) mainly due to the small quantity of labeled data, especially for the Colombian conflict dataset. Thus, we plan to distribute the labeling processing to several experts aiming to build a dataset from multiple annotators. Hence,

the objective is to use transfer learning with pre-trained models based on deep learning such as LSTM or attention models combined with learning from crowds layers as in [23,62], which can favor the semantic orientation representation.

**Author Contributions:** Conceptualization, J.G.-G. and P.R.-C.; data curation, J.P.-V.; methodology, P.R.-C., J.G.-G. and M.R.-H.; project administration, M.R.-H. and P.R.-C.; supervision, J.G.-G. and J.P.-V.; resources, P.R.-C. and J.P.-V. All authors have read and agreed to the published version of the manuscript.

**Funding:** Under grants provided by the "Fondo nacional de financiamiento para la ciencia, la tecnología y la innovación Francisco José de Caldas", the "Ministerio de ciencia, tecnología e innovación"—(MinCiencias), the "Centro nacional de memoria histórica", the "Universidad Católica de Pereira", and the "Universidad del Quindío" under the project: "Analytical Center of University Cultural Productions in the Context of the Conflict (caPAZ)."—code 1349-872-76354, agreement 872 of 2020.

**Institutional Review Board Statement:** Institutional review is not applicable because this work does not include data involving humans or animals.

**Informed Consent Statement:** This study does not involve humans; accordingly, informed consent is not required.

**Data Availability Statement:** Datasets are publicly available at: http://www.lsi.us.es/~fermin/corpusCine.zip (accessed on 31 March 2021), https://archive.ics.uci.edu/ml/datasets/Paper+Reviews (accessed on 31 March 2021), and https://zenodo.org/record/6384840#.Yj5Av-fMJPZ (accessed on 31 March 2021).

**Conflicts of Interest:** The authors declare that this research was conducted without any commercial or financial relationships that could be construed as a potential conflict of interest.

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
