# Peer review of "Self-Supervised Sentiment Analysis in Spanish to Understand the University Narrative of the Colombian Conflict"

_applsci, doi:10.3390/app12115472_

Round 1

Reviewer 1 Report

The authors propose a self-supervised approach for sentiment analysis in Spanish that comprises a lexicon-based method and a supervised classifier.

I have a couple of major concerns regarding the approach. 

From lines, 61-72, authors describe how to overcome challenges in a limited data scenario. Another approach that the community has taken is to perform transfer learning. That is to train a machine learning model on a data-rich scenario and then perform transfer learning. I would like to see a discussion about it too. 

I am not confident about the machine translation step. I understand that it is vital to perform machine translation to get some text in Spanish. However, as we all know, machine translation is not the best. As a result, a bias in the machine translation system can appear in the dataset too. I would like to see a quality evaluation involving machine translation quality estimation from the authors. 

What are the reasons for the selection of machine learning classifiers? Why did not you use state-of-the-art text classification algorithms such as LSTMs and transformers? I would like to see a discussion about adding a strong machine learning classifier. 

Overall, I have happy about the language. However, there are several mistakes and would be better if the manuscript could be proofread by a native speaker. 

  1. I would not use the word "face" in line 61. Tackle would be a better option. 
  2.  Line 259 -  In conclusion, a total of 2373 digital news-articles were analyzed. The information is available in Table 2. 

Reviewer 2 Report

The article proposes a hybrid approach combining a lexicon-based approach with a supervised method for polarity prediction on a Spanish dataset after translation. The article is easy to read and follow; however, it has limited to no-novelty except for the translation part compared to the previous work reported in [13]. 

As the only significant addition is the Google translation, I wonder what impact the model has on the sentiment classification and what words affect the classification accuracy due to translation. The authors in https://ieeexplore.ieee.org/abstract/document/9529190 reported that certain words often lose the exact context and meaning, resulting in a polarity shift as a result of translation. I suggest authors present an analysis of the impact of Google translation from Spanish to English for sentiment classification. 

There is also no SOTA comparison to embedding models. The ones presented in the article are quite primitive. Many advanced attention-based deep learning models have been proposed in recent years. I doubt if Spanish is a low-resource language, and you may find many pre-trained models for it. Have authors given a thought about it?

I also see a lot of similarities between your proposed pseudo-labeling approach to the weakly supervised approach for sentiment classification presented in https://ieeexplore.ieee.org/abstract/document/9110884.  

As a future work, I suggest authors look into improving classification accuracy on highly imbalanced text datasets using text generation techniques for natural language models as a data augmentation technique, in addition to 46. 

Other minor comments:

  1. Use one; "Accordingly, Nevertheless, we ..." on line 131
  2. Typo on line 320 - "fo both..."

Round 2

Reviewer 1 Report

The authors have addressed all of my comments.

Reviewer 2 Report

The authors have addressed my comments.